# Protein Backbone and Average Particle Dynamics in Reconstituted Discoidal and Spherical HDL Probed by Hydrogen Deuterium Exchange and Elastic Incoherent Neutron Scattering

**DOI:** 10.3390/biom10010121

**Published:** 2020-01-10

**Authors:** Valentin Gogonea, Judith Peters, Gary S. Gerstenecker, Celalettin Topbas, Liming Hou, Jérôme Combet, Joseph A. DiDonato, Jonathan D. Smith, Kerry-Anne Rye, Stanley L. Hazen

**Affiliations:** 1Department of Cardiovascular and Metabolic Sciences, Lerner Research Institute, Cleveland Clinic, Cleveland, OH 44195, USA; gsgers@hotmail.com (G.S.G.); ctopbas1@gmail.com (C.T.); didonaj@ccf.org (J.A.D.); smithj4@ccf.org (J.D.S.); hazens@ccf.org (S.L.H.); 2Center for Microbiome and Human Health, Cleveland Clinic, Cleveland, OH 44195, USA; 3Department of Chemistry, Cleveland State University, Cleveland, OH 44115, USA; 4Institut Laue Langevin, 71 avenue des Martyrs, CS 20156, F-38042 Grenoble CEDEX 9, France; jpeters@ill.fr (J.P.); jerome.combet@ics-cnrs.unistra.fr (J.C.); 5Université Joseph Fourier Grenoble I, UFR PhITEM, F-38041 Grenoble CEDEX 9, France; 6Lipid Research Group, School of Medical Sciences, Faculty of Medicine, University of New South Wales Sydney, Sydney NSW 2052, Australia; liming.hou@mq.edu.au (L.H.); k.rye@unsw.edu.au (K.-A.R.); 7Institut Charles Sadron, CNRS-Université de Strasbourg, 23 rue du Loess, B.P. 84047, 67034 Strasbourg CEDEX 2, France; 8Department of Cardiovascular Medicine, Cleveland Clinic, Cleveland, OH 44195, USA

**Keywords:** apolipoprotein A-I, high-density lipoprotein, chemical cross-linking, MS cross-linking, hydrogen-deuterium exchange mass spectrometry, elastic incoherent neutron scattering, thermal neutron, lipoprotein dynamics, lipoprotein softness, protein backbone dynamics

## Abstract

Lipoproteins are supramolecular assemblies of proteins and lipids with dynamic characteristics critically linked to their biological functions as plasma lipid transporters and lipid exchangers. Among them, spherical high-density lipoproteins are the most abundant forms of high-density lipoprotein (HDL) in human plasma, active participants in reverse cholesterol transport, and associated with reduced development of atherosclerosis. Here, we employed elastic incoherent neutron scattering (EINS) and hydrogen-deuterium exchange mass spectrometry (HDX-MS) to determine the average particle dynamics and protein backbone local mobility of physiologically competent discoidal and spherical HDL particles reconstituted with human apolipoprotein A-I (apoA-I). Our EINS measurements indicated that discoidal HDL was more dynamic than spherical HDL at ambient temperatures, in agreement with their lipid-protein composition. Combining small-angle neutron scattering (SANS) with contrast variation and MS cross-linking, we showed earlier that the most likely organization of the three apolipoprotein A-I (apoA-I) chains in spherical HDL is a combination of a hairpin monomer and a helical antiparallel dimer. Here, we corroborated those findings with kinetic studies, employing hydrogen-deuterium exchange mass spectrometry (HDX-MS). Many overlapping apoA-I digested peptides exhibited bimodal HDX kinetics behavior, suggesting that apoA-I regions with the same amino acid composition located on different apoA-I chains had different conformations and/or interaction environments.

## 1. Introduction

Knowing the structure and dynamics of biomolecules is crucial for understanding their physiological functions. Molecules, in general, have distinct dynamic behavior according to their atomic composition, and the pattern of their intermolecular interactions determines both the dynamics of their components and their average softness and flexibility. For example, lipids are generally more dynamic (softer and more flexible) and experience high thermal fluctuations in atomic motion at ambient temperature because the majority of their intermolecular interactions are weak (van der Waals), whereas proteins and DNA are more rigid due to the presence of additional stronger intermolecular interactions like hydrogen (H)-bonds, charge-charge, charge-dipole, and dipole-dipole interactions [1].

A powerful spectroscopic technique used to interrogate the softness and flexibility of biomolecules is elastic incoherent neutron scattering (EINS, performed on neutron spectrometers) in which the variation in momentum of no-energy-exchange scattered neutrons is detected and linked to averaged motion (mean square displacement, MSD) of hydrogen atoms in biomolecules because hydrogen atoms have high incoherent neutron scattering cross-section compared to other atoms present in biological systems [2]. Averaged MSD can be further related to average force constants that translate into the stiffness or resilience properties of biomolecules [3]. The force constant obtained from EINS measurements can be interpreted as the average force constant of virtual springs in between groups of atoms, depicting the interactions, analogous to how normal (vibrational) modes of collective motions in polyatomic molecules are characterized through force constants (which relate to the second derivative of the potential energy with respect to atomic displacements). Lipid vesicles, cell membranes, and lipoproteins exhibit a wide variety of dynamical behaviors dictated by their size and composition, and their softness and flexibility have some bearing on their physiological functions. For example, highly dynamic lipoproteins make excellent lipid transporters in the circulatory system. Lipid-containing biomolecules constitute attractive targets for investigation by EINS because their softness and flexibility impact their biological functions, as shown in earlier studies on 1,2-dimyristoyl-*sn*-glycero-3-phosphocholine (DMPC) vesicles [4], very-low-density lipoproteins (VLDL), and low-density lipoproteins (LDL) [5,6,7].

The present study looked at average particle softness and flexibility, and protein backbone dynamics of two reconstituted functional lipoproteins: discoidal and spherical high-density lipoproteins (HDL), by using a combination of EINS and hydrogen-deuterium exchange mass spectrometry (HDX-MS) analyses. Plasma levels of HDL cholesterol are inversely associated with the risk of heart disease [8,9,10], and significant evidence supports the role of HDL particles as lipid transporters in reverse cholesterol transport (RCT), where the lipoprotein provides the means for cholesterol to be shuttled from peripheral tissues to the liver and steroidogenic tissues [11,12,13]. Vital to the understanding of HDL physiological function is both the structure and the dynamics of apolipoprotein A-I (apoA-I) bound to lipids, and the plasticity of the lipoprotein particle as a whole. Despite the fact that the constituent proteins of HDL found in human plasma are extremely heterogeneous, HDL can be categorized into two primary groups, those that contain apoA-I but not apolipoprotein A-II (apoA-II) and those that contain apoA-I as well as apoA-II [14]. Since apoA-I represents approximately 70% of the protein content of HDL by mass, it is an important factor influencing its biological activity [15,16].

Due to the difficulty in isolating homogenous HDL preparations from plasma, for years, a majority of structural studies of HDL have focused on reconstituted discoidal HDL (rHDL), which consists of two molecules of apoA-I, phospholipid, and free cholesterol [15,16]. The existing structural models of rHDL have two apoA-I chains in an anti-parallel orientation that exist either as a double belt shape around the circumference of a central lipid bilayer comprised of phospholipid and free cholesterol [15,16,17,18,19,20,21,22,23] or as a superhelical shape wrapping around a predominantly micellar lipid core [23,24,25]. Spherical HDL (sHDL), on the other hand, is the mature form of HDL that transports cholesterol back to the liver that has typically been converted into cholesteryl ester in peripheral tissues and has a lipid core made mostly of cholesteryl esters surrounded by a phospholipid monolayer in which apoA-I is embedded, the latter acting as a scaffold, offering structural integrity to the particle.

Only a few studies have addressed experimentally the structure of sHDL in detail due to its compositional heterogeneity [14,23,26,27,28,29,30]. While some characterization studies suggest that apoA-I in reconstituted rHDL and sHDL has similar α-helical content [31] and that the overall structure of apoA-I is similar in sHDL, either reconstituted from individual constituents or isolated from human plasma [31,32], others have noted changes in particle’s properties that infer the presence of structural differences between apoA-I in rHDL versus sHDL [18,24,25,30,33,34,35,36,37]. For example, previous mass spectrometry-based studies, utilizing chemical cross-linking of rHDL [18,38], sHDL [29,30], and HDL isolated from human plasma [39], have demonstrated similar apoA-I inter-chain cross-links, suggesting that apoA-I adopts a relatively common architecture in HDL regardless of particle size and protein content [18,29,30,38,39]. This information, combined with studies involving HDX-MS on rHDL [18,40] and relatively heterogeneous sHDL preparations separated from human plasma [31], demonstrate the need for structural knowledge derived from HDX-MS on relatively homogenous sHDL preparations to resolve how the observed cross-linking pattern remains largely the same, while the shape of the particles changes drastically [24,30].

In this work, we utilized a combination of biophysical approaches, including HDX-MS [41,42,43,44,45], EINS, and MS cross-linking studies, to examine the secondary structure, conformational diversity, and the local dynamics of apoA-I in reconstituted sHDL particles that are relatively homogenous populations of physiologically functional particles similar in shape to the more heterogenous sHDL in plasma. The structural and dynamical information obtained through these analyses helped understanding how the protein-protein interaction among apoA-I chains, their dynamics, and their interaction with the particle’s lipid core contributed to HDL remodeling during its life cycle, and HDL interaction with plasma enzymes (lecithin cholesteryl acyltransferase (LCAT), paraoxonase 1 (PON1), cholesteryl ester transfer protein (CETP)) and cell receptors (ATP binding cassette transporter (ABCA1), ATP binding cassette sub-family G member 1 (ABCG1), the scavenger receptor class B type 1 (SR-BI)).

## 2. Materials and Methods

### 2.1. Human Studies

All participants gave written informed consent, and all study protocols were approved by the Institutional Review Boards of the Cleveland Clinic.

### 2.2. HDL Preparation

Human apolipoprotein A-I was purified from units of plasma recovered from healthy volunteers, as previously described [18]. ApoA-I was isolated from HDL recovered from fresh human plasma by using sequential ultracentrifugation in the 1.07–1.21 g/mL density range with KBr. HDL proteins were precipitated with methanol/chloroform/cold water-washed ether before re-suspension (20 mM Tris, pH 8.5, 6 M urea). Human apoA-I was purified by ion-exchange chromatography (Q sepharose HP HiLoad 26/10 column from GE Healthcare, Pittsburg, PA, USA) and stored in PBS with 3M guanidine HCl at −80 °C under argon blanket until use.

Nascent (discoidal) HDL particles (rHDL) were reconstituted by the cholate dialysis method [32], using an initial ratio of 100:10:1, 1-palmitoyl-2-oleoyl-*sn*-glycero-3-phosphatidylcholine (POPC): cholesterol: apoA-I (mol:mol:mol), and purified by gel filtration chromatography (Appendix A) [18]. Reconstituted spherical HDL particles (rsHDL) were prepared by incubating rHDL with isolated human lecithin cholesteryl acyltransferase (LCAT) and human low-density lipoprotein (LDL) at 37 °C for 24 h, as previously described [32,46,47]. Reconstituted sHDL particles were separated by sequential ultracentrifugation in the density range of 1.07–1.21 g/mL. The rHDL and rsHDL preparations were purified by size exclusion chromatography (Sephacryl S-300 column, GE Healthcare Pittsburg, PA, USA), and their purity was determined using 4–20% native gel electrophoresis (Appendix A). The size of the rHDL and rsHDL particles was determined by comparison with protein standards with known Stokes diameter (GE Healthcare Pittsburg, PA, USA) and by dynamic light scattering using a particle size analyzer (NanoPlus-1, Particulate Systems, Norcross, GA, USA).

### 2.3. Compositional Analysis of Reconstituted HDL

The protein and lipid content of rHDL and rsHDL particles were determined, as described previously [18,24]. In brief, the phospholipid composition of HDL particles was determined by micro-phosphorus assay [18,24], and the amount of protein was quantified using a modified Lowry protein assay and confirmed by stable isotope-dilution LC/MS/MS analysis of Phe, Tyr, Lys, using the protein sequence of human apoA-I. Calculated protein mass from measured amino acids agreed within ±2%. The free and total cholesterol contents in rsHDL preparations (following KOH saponification) were determined enzymatically (Stanbio Laboratory and Wako Diagnostics kits, Richmond, VA, USA). Cholesteryl ester in sHDL samples was determined by subtracting free cholesterol from total cholesterol. Chemical cross-linking of rHDL and rsHDL was performed with bis(sulfosuccinimidyl)suberate (BS3) cross-linker at various concentrations at 25 °C for one hour, as described in [18,24] (Supplementary Figures S1B and S2B).

### 2.4. Sample Preparation for Elastic Incoherent Neutron Scattering

Elastic incoherent neutron scattering requires a high amount of sample (~100 mg) and a lengthy measuring time (hours) for each sample. We used samples of ~0.5 mL (200 mg/mL of HDL) in a phosphate buffer solution (PBS). The samples were first lyophilized and then rehydrated by vapor exchange over D_2_O at ambient temperature in a desiccator until final D_2_O content was 0.49 *g*/*g* (known to correspond to at least one full hydration layer in proteins, [48]). The HDL samples were placed in flat aluminum sample holders of dimensions 30 × 40 × 0.5 mm^3^ and sealed with indium. Samples were weighed before and after the neutron scattering to check that the mass, hydration, and morphology did not change, and no alterations in sample homogeneity were observed.

### 2.5. Elastic Incoherent Neutron Scattering of Reconstituted HDL Particles

Two spectrometers (IN13 [49] and IN16 [50]) at Institut Laue-Langevin (ILL), Grenoble, France [3] were used for scattering. IN13 is a thermal backscattering spectrometer with an energy resolution ΔE of ~8 μeV (~100 ps time window), whereas IN16 is a cold neutron backscattering spectrometer with an energy resolution ΔE of ~0.9 μeV (~1 ns time window). The instruments differ furthermore in the range of momentum transfer *Q*, thus accessing different ranges of protein and lipid atoms motion amplitudes. Both spectrometers are sensitive to the type of motions the protein and lipid components of HDL experience, but the larger time window of IN16 allows also the detection of larger motions, typically of subdomains in the sample structure [51].

Neutron scattering has both coherent and incoherent components; the coherent component renders structural details and collective excitations, while the incoherent one is characteristic to the average motion of individual atoms. Since the incoherent scattering cross-section of the hydrogen atom is the highest in these lipoprotein samples, the motion of hydrogen atoms, or the movement of molecular groups they are attached to, was detected by EINS. To avoid interference from the movement of water hydrogen atoms in the hydration shell of particles, D_2_O was used to hydrate the dry samples, which singled out the hydrogen atoms in the biological particle. On IN13, the temperature scan between 20 and 160 K was performed with steps of 20 K, between 170 and 280 K with steps of 10 K, and steps of 5 K between 285 and 305 K. On IN16, a continuous temperature ramp of 0.3 K/min was used, and the data were binned to intervals of 5 K.

The EINS intensity is given by the following Gaussian approximation [52]:(1)$$I_{el}\left(Q,\omega =0\pm \Delta E\right)\approx I_{0}exp\left(-\frac{1}{6}\langle u^{2}\rangle Q^{2}\right)$$
where <*u*^2^> is the average atomic mean square displacement (MSD). For *Q* → 0, the approximation is strictly valid, and it holds up to <*u*^2^> *Q*^2^ ≈ 2. On IN13, we identified the momentum transfer domain: 0.52 Å^−1^ < *Q* < 2.06 Å^−1^ to fulfill this approximation (where the logarithm of the intensity is approximately linear with *Q*^2^, even for temperatures close to ambient). On IN16, the range 0.54 Å^−1^ < *Q* < 1.85 Å^−1^ was very similar to the range on IN13, which allowed the comparison of the same amplitudes on different time scales. The MSD was obtained for each temperature from the slope of the semi-logarithmic plot of the incoherent scattering function through the following formula:(2)$$\langle u^{2}\rangle \approx -6\frac{d\ln I_{el}\left(Q,\omega =0\pm \Delta E\right)}{dQ^{2}}$$

The scattering intensity from the empty sample holder was subtracted, and the normalization was done with respect to vanadium, a completely incoherent scatterer [53]. Correction of the absorption was done with the Large Array Manipulation Program (LAMP) from ILL, used for neutron scattering data treatment [54], based on the correction formula of Paalman–Pings coefficients [55].

The MSD represented the average molecular flexibility at a given temperature due to the thermal energy present in the sample. Moreover, information about the resilience (stiffness) of the sample in a given temperature range could be obtained, based on the temperature dependence of MSD [1]; the effective force constant <*k*> in the samples could be calculated [1] as:(3)$$\langle k\rangle =\frac{0.00276}{d\langle u^{2}\rangle /dT}$$

Here, <*k*> is expressed in N/m, when <*u*> is given in Å^2^ and *T* (the absolute temperature) in Kelvin.

### 2.6. HDX-MS of Reconstituted Spherical HDL Particles

Hydrogen-deuterium exchange (HDX) in reconstituted sHDL (rsHDL) samples was obtained at 0 °C in an ice/water bath at pH = 6.8 (pD = 7.2) in D_2_O-PBS for various time intervals: 30, 100, 300, 1000, 3000, and 10,000 s. All experimental solutions [rsHDL preparations, D_2_O buffer (10 mM sodium phosphate, 150 mM sodium chloride, pH 6.8 (pD 7.2)), hydrochloric acid (HCl), and proteases (pepsin and fungal protease XIII)] were allowed to reach the bath temperature, and the HDX was conducted by diluting rsHDL with D_2_O buffer to yield a final protein concentration of 200 µg/mL. The HDX was quenched by acidification with HCl, lowering the pH to 2. The resulting solution was subjected to rapid proteolytic digestion (pH = 2, enzyme conc. = 20 µg/mL) using either pepsin (digestion time = 5 min) or fungal protease XIII (8 min) (Appendix A, peptide coverage: protease XIII = red, pepsin = green) and injected into the HPLC-MS system (a binary LC20 UPLC pumps, Shimadzu, Kyoto, Japan, and an LTQ ion-trap mass spectrometer, Thermo Scientific, Waltham, MA, USA). The gradient for peptides elution started at 200 μL/min (0.5% formic acid aqueous solution) for 1 min, decreased to 50 μL/min (10% acetonitrile, 0.5% formic acid), and continued (50% acetonitrile, 0.5% formic acid) for 10 min. An Everest 238CV C18 guard column (2.1 mm diameter, Grace, Deerfield, IL, USA) was used as trap column, and a Vydac 238TP C18 guard column (2.1 mm diameter, Grace, Deerfield, IL, USA) was used as the analytical column. Control samples (80% deuteration and 0% deuteration) were analyzed to measure back exchange (D→H) and to establish retention times from the MS spectra of apoA-I peptides, respectively. The MS/MS spectra of the non-deuterated apoA-I peptides were searched utilizing the SEQUEST algorithm (Thermo Proteome Discoverer version 1.1) and confirmed manually. The deuterated spectra were manually interpreted.

### 2.7. Analysis of HDX-MS Spectra

Deuterium (D) incorporation (DI) was determined by comparing the mass difference between the centroid *m*/*z* value of a specific deuterated peptide’s isotopic envelop with the centroid *m*/*z* for the non-deuterated peptide isotopic envelope. This value was adjusted to account for the charge state of the peptide ion and to correct for back exchange in the overall procedure, which, in general, remained at less than 21 percent (12% on average). While analyzing the MS spectra of deuterated peptides, we noticed that for certain peptides, the MS envelop and the centroid *m*/*z* value changed as a function of HDX time. This phenomenon was earlier observed and coined as bimodal HDX kinetics [31] and is due to the existence of the same protein domain in either dissimilar conformations or within distinct interaction environments. On the other hand, any peptide can have residues that exchange fast or slow (i.e., have different HDX rate constants), a behavior coined biphasic HDX kinetics. Peptides that exhibit either bimodal or monomodal HDX kinetics can also have biphasic kinetics [56]. Bimodal HDX kinetics was detected in more than half of the digested peptides, and most peptides exhibited biphasic HDX (vide infra).

### 2.8. Kinetic Analysis of HDX-MS Data

The deuterium incorporation (DI) values obtained from the HDX-MS experiments were processed in two steps. First, the HDX reaction conditions (pH_rxn_ = 6.8 (pD = 7.2), pH_quench_ = 2 (quenching) (pD = 2.4), temperature (T) = 0 °C) and the amino acid composition of each peptide identified by the MS/MS experiment were used to determine the HDX kinetic rate constants for individual residues in random coil conformation. This step was accomplished by using the program DEXANAL (version 6.8) developed in our lab [25,37] that implements the methodology for HDX kinetics of random coil peptides developed by Bai et al. [57].

The analysis of the mass spectra of deuterated peptides obtained by using collision-induced ionization did not allow the identification of individual HDX sites within the peptide, but rather the total number of D incorporated in the peptide. Hence, only average HDX rate constants for peptide domains that exchange with approximately the same rate could be estimated. The average rate constants could be determined by using stretched exponential functions to fit DI values. For example, to facilitate this kind of analysis, average HDX rate constants for the random coil peptides (Appendix A) were estimated by fitting DI values for each peptide to a single stretched exponential function (Appendix A).

Fitting the experimental DI values for the apoA-I peptides, as a function of the HDX time to stretched exponential functions, produced average HDX rate constants for the slow or fast exchanging domains (biphasic HDX kinetics) of each digested peptide in native conformation (as opposed to HDX rate constants for individual residues in random coil conformation) (Appendix A).

The HDX protection factors (Pf) for peptide domains, exhibiting biphasic HDX kinetics (Appendix A), were calculated as the ratios of the HDX rate constants for the slow or fast exchanging domains and the HDX rate constants for the same domains in random coil conformation [57]. The Pf for all digested peptides identified by MS are listed in Appendix A.

HDX protection factors (Pf) for individual amino acid residues in peptides in the native state were predicted from peptide overlap in the following way: all digested peptides were grouped into two sets based on their relative abundances (Appendix A). Peptides displaying bimodal HDX with relative abundance one were assigned to a peptide pool assumed to originate from an apoA-I dimer, while peptides having a relative abundance <1 were incorporated into a peptide pool assumed to originate from an apoA-I monomer. The peptides displaying unimodal HDX were assigned to both pools. This peptide grouping was consistent with an rsHDL particle in which the three apoA-I chains are organized in a dimer/monomer configuration (the helical dimer hairpin monomer (HdHp) model) [30].

To predict HDX protection factors for individual residues in a peptide in its native state, a group of overlapping peptides was first aligned, and the Pf for individual residues was calculated using the HDX rate constants for either the fast or slow HDX sites (for peptides displaying biphasic HDX) and the HDX rate constants for individual residues in random coiled state. After alignment, the HDX protection factors were randomly assigned to individual residues in each of the overlapping peptides and permuted iteratively until the consistency was reached in the Pf assignment for all overlapping peptides (that is, aligned residues in the overlapping peptides set have the same Pf).

## 3. Results and Discussion

Like other macromolecular assemblies (i.e., protein complexes, other lipoproteins), the physiological roles of HDL can be diverse [58], and its ability to execute these functions depends on both its overall softness and the mobility of various regions of the particle (e.g., ability to expose or hide certain protein domains or change overall shape). These two facets of particle dynamics, i.e., the overall softness of the particle and its local average flexibility, are not usually discussed congruently, so we fathomed to bring together the results on both the local and the overall dynamics of HDL and discuss their impact on its physiological functions in a coherent fashion. In lipoproteins, local flexibility in protein components facilitated interactions with plasma enzymes (e.g., LCAT, PON1, CETP, myeloperoxidase (MPO), etc.) and cell receptors (e.g., SR-BI, etc.), while the overall softness/flexibility of the particle could be critical for activating/inhibiting physiologically relevant regions of the particle, or migration through tissue/vasculature in general. For example, HDL has two regions where the LCAT enzyme (that matures HDL) can bind, but only one seems to be physiologically active, probably a consequence of the cooperative movement between the lipid core and the protein, which produces asymmetric topographies around the two LCAT active sites, thus rendering one site inactive [25].

While both EINS and HDX methods produced information about particle dynamics, they did not corroborate each other findings but were complementary in the sense that they extracted different information about the particle’s dynamics. Thus, it was not possible to infer from HDX data the force constant for the overall dynamics obtained from the EINS measurements, or rate constants and protection factors for HDX from EINS measurements. Both EINS and HDX methods are well-established techniques to explore different facets of particle dynamics [1,59]. EINS can be applied to study any macromolecular assembly [1], while HDX is limited to proteins because the backbone amide H experiences a slow exchange with D, which can be captured within the time frame of the experiment [59].

### 3.1. Reconstituted HDL Particles Have Composition and Biological Activity Similar to Plasma HDL

Before proceeding with neutron scattering (EINS) and hydrogen-deuterium exchange (HDX-MS) measurements on HDL preparations, the particles were extensively characterized to quantify composition and ensure both homogeneity and appropriate biological function. Discoidal and spherical HDL particles were reconstituted, purified, and characterized, as described in the Materials and Methods section. The native gradient gel of the rHDL preparation (Appendix A) showed a particle with a Stokes diameter of 9.6 nm, similar with rHDL particles we prepared and used in earlier studies, and also reported by many other researchers in the field [24]. The native gradient gel of the rsHDL preparation showed a particle with a Stokes diameter of 8.8 nm ([30], Appendix A). The chemical composition of the rHDL particle gave a molar ratio of 86 ± 5.1/9 ± 1.5/1 POPC/unesterified cholesterol/apoA-I, and for the rsHDL particle, the molar ratio was 28 ± 3.2/3 ± 0.8/16 ± 2.6/1 of POPC/unesterified cholesterol/cholesteryl ester/apoA-I. The SDS-PAGE gel of BS3 cross-linked rHDL (Appendix A) at ratios of rHDL:BS3 of 1:20 and 1:200 displayed mostly a band of apoA-I dimer, while for rsHDL, the SDS-PAGE gel, at ratios of sHDL:BS3 of 1:2, 1:20, and 1:200, showed monomer, dimer, and trimer (Appendix A). As previously shown with similar rHDL and rsHDL preparations we made and studied using HDX-MS and small-angle neutron scattering (SANS) [18,24,30], preparations made and examined in the current HDX studies also displayed typical biological activity, including cholesterol efflux activity when incubated with cholesterol loaded macrophages, and specific binding to the SR-BI receptor [18,24,30]. Thus, the rHDL and rsHDL particles used in this study were mono-disperse and homogeneous, with lipid/protein composition, particle size, and biological function similar to plasma HDL.

We noted that, compared to rHDL of 9.6 nm used in this study (with two apoA-I molecules and ~10% cholesterol in the lipid phase), the nascent HDL particles produced by the ABCA1 receptor in HEK cells were diverse in size and lipid/protein composition. Despite these differences in composition, reconstituted discoidal HDL (rHDL) proved to be an effective therapeutic agent capable of improving reverse cholesterol transport in humans [60,61,62], which means rHDL can function similarly to nascent HDL produced in living organisms.

### 3.2. Elastic Incoherent Neutron Scattering of Reconstituted HDL Particles Reveals that Discoidal HDL is Softer and More Flexible than Spherical HDL at Ambient Temperature

The concept of molecular flexibility relates to the amplitudes of atomic motions, while resilience is a material property that describes the stiffness or hardness of the molecular structure to deformation in response to external forces. Elastic incoherent neutron scattering measures the average displacement over all atoms (MSD), so the atomic motion is interpreted in terms of virtual interactions (spring-like) that can be, in the case of a lipoprotein, between protein domains, protein and lipid domains, and lipid domains [1]. The notion of force constants of “virtual springs” relates to average forces that the macromolecular assembly experiences due to the oscillatory movement it performs (as a consequence of thermal fluctuations). We could draw a parallel for this interpretation of the force constant with multiatomic molecules where normal (vibrational) modes of groups of atoms, which are not necessarily bound together, vibrate collectively, and their frequency of vibration can be related to a “force constant” associated with this collective periodic motion, which is thought of as the motion of a virtual string.

Neutron scattering experiments on rHDL hydrated powder were performed on the IN13 instrument only, while measurements of rsHDL hydrated powder were performed on both IN13 and IN16. Hydrated powders of biological samples are not exactly in the same state as biomolecules in cells or other native environments, which are mostly characterized by hydration above the ones in powders; thus, not all aspects of picosecond dynamics are accurately described. Although certain fast processes of few picoseconds are only activated in the presence of a hydration level beyond one surface layer, the dynamical transition, in which we were interested here, is present in hydrated powder samples [63]. Comparisons of two samples prepared in the same conditions are therefore significant. Note that lyophilization of the lipoprotein samples and replacement with D_2_O prior to EINS might change the shape of the lipoproteins, but not their composition. However, no morphological or compositional changes in the lipoprotein samples after lyophilization or EINS measurement were observed. Therefore, such treatment of the sample did not alter the structure/composition of the samples. Nevertheless, the effect of lyophilization on biological systems was investigated by J. Perez et al. [64], who concluded that complete coverage of lyophilized sample with a solvation layer at 0.4 g/g generated hydrogen-bonding pathways that allowed surface residues to diffuse locally, and further hydration only increased diffusion.

Figure 1 shows the mean square displacement (MSD) of rHDL and rsHDL, in hydrated powder form, measured on IN13 as a function of absolute temperature. The spherical particles were clearly more flexible at low temperatures, and, after a crossover at around 250 K, the discoidal particles were much more flexible at a higher temperature. Results for sHDL, in powder form, measured for similar ranges of momentum transfer (*Q*) on IN13 and IN16 (Appendix A) showed likewise changes in MSD with respect to absolute temperature. However, the error bars for data collected on IN16 were smaller because the neutron flux on this instrument was higher than on IN13. In addition, the MSD was slightly larger on IN16 because all motions within a time-window up to 1 ns were detected as, for instance, movements of subdomains, in addition to the localized motions due to individual atoms or group of atoms. For instance, the MSD was 2.1 Å^2^ at 305 K for the rsHDL sample measured on IN13, and 2.4 Å^2^ at 304.6 K when measured on IN16. The crossover of the data found around 250 K, which would mean that at low (non-physiological) temperatures, the local motions with shorter timescale (100 ps) were predominant, whereas, at higher temperatures, the larger amplitudes on nanosecond timescale dominated.

While no prior EINS studies of HDL have been reported, our results were consistent with earlier EINS measurements on plasma VLDL and LDL [5,6], indicating that lipoproteins are softer and much more flexible than well-studied globular proteins, such as myoglobin or lysozyme [1,65]. Moreover, our results revealed a surprising similarity between the MSD curves of rHDL/rsHDL (Figure 1) and VLDL/LDL, published previously by Mikl et al. (Figure 1 in [5]). The MSD for both lipoproteins pairs was about the same up to ~270 K, after which the MSD curves started to diverge. They presented three linear domains that changed their slope at ~200 and ~250 K. The various slopes could be interpreted in terms of average force constants [1], maintaining the biological structure in the sample (Table 1). In agreement with the findings for the MSD, the average force constants for the molecular motion of rHDL and rsHDL, given in Table 1, indicated that at temperatures lower than 200 K, rsHDL was slightly more dynamic than rHDL.

In principle, based on the types of intermolecular interactions within lipids (dispersion) and proteins (Coulomb, H-bond, dispersion), we would expect the lipid domain of HDL to be more dynamic than the protein; but at very lower temperatures (< 200 K), we found that the protein motion dominated the lipid motion in the average particle dynamics, in agreement with earlier findings on natural membranes [66] and purple membranes [67], while at high temperatures (>200 K), the internal dynamics of rHDL exceeded that of rsHDL (given that rHDL had a higher lipid to protein ratio than rsHDL), in agreement with previous results published on other lipoproteins (VLDL and LDL [5,6,7], natural membranes [66], and purple membranes [67]).

The inflection points in Figure 1 were chosen somehow arbitrarily according to the variations in the slope, but they allowed to show that the inversion in the tendency between the two samples occurred only at the highest temperature. The first inflection point was close to the dynamical transition temperature (200–220 K), but it depended on the instrument resolution and had never been defined exactly. The second inflection point was close to a transition, which has also been observed for purple membranes [1], and which might be typical for systems containing lipids. While the inflection points indicate clear changes in the dynamic behavior of HDL, it is not possible to attribute specific kinds of motions to the three ranges of temperature (Figure 1). According to Frauenfelder’s interpretation in terms of conformational sub-states [68], one can only conclude that the motions at low temperatures are harmonic, while at high temperatures, they are an-harmonic.

Note that lipid phase transitions are observed in the dynamics of pure lipid samples [66], but are hardly visible in mixed systems, as in the present case [4]. Thus, during the gradual change in temperature, we did not register typical phase transitions for POPC or cholesteryl ester core that occurred at ~270 K and ~306 K, respectively. This was probably because the subtle changes in lipid structure that accompanied these phase transitions (i.e., lipid interdigitation, lipid tail-tilt angle, hydrocarbon trans-gauche isomerization) did not alter the dynamics of the rsHDL and rHDL particles according to EINS, which did not monitor thermal changes associated with phase transitions. Figure 1 shows a continuous monotonic change in MSD without slope alteration from 250 K to 300 K, but we could not rule out that changes in the MSD slopes at ~200 K and ~250 K were associated with phase transitions in the lipid-protein assemble.

A recent study reported that plasma HDL containing apoA-I/apoA-II participated less in lipoprotein remodeling than HDL containing apoA-I only and that there was little difference between the metabolic fate of HDL particles with or without apoA-II [69]. We hypothesized that the dynamics of HDL particles containing either apoA-I or apoA-I/apoA-II should be similar if their lipid/protein ratio is the same. The difference in the ease with which these distinct lipoproteins remodel likely stems from changes in interaction with plasma enzymes and cell receptors due to the presence of apoA-II, rather than differences in overall dynamic behavior.

### 3.3. Bimodal HDX Kinetics in Reconstituted Spherical HDL Indicates Conformational Diversity and/or Distinct Molecular Environments for ApoA-I Chains

Our HDX-MS analysis of rsHDL used a combination of acid-active proteases for digestion to enhance apoA-I coverage and produced over 150 overlapping peptides (Appendix A). With extensive (near-complete) coverage and multiple overlapping peptides, it was possible to comprehensively utilize the kinetic behavior of all overlapping peptides to predict probabilities in HDX behavior across the apoA-I polypeptide chains in rsHDL with improved resolution. This was done with the program DEXANAL, as described in the Materials and Methods section. The bimodal HDX kinetic behavior of apoA-I peptides in rsHDL is exemplified in Figure 2, where the progression of the mass spectra as a function of HDX time for apoA-I peptides E_212_-L_222_ (EDLRQGLLPVL) and Y_115_-L_126_ (YRQKVEPLRAEL) are shown.

Cartoons of the deconvoluted isotopic envelopes, predicted from each peptide, representing distinct modes of DI, were drawn to demonstrate the phenomenon of bimodal HDX kinetics. A fast exchanging domain of apoA-I (E_212_–L_222_, red peak in Figure 2 left) could be observed of approximately half the intensity of a slower primary isotopic envelope (blue peak in Figure 2 left). In the case of apoA-I domain Y_115_–L_126_, the inverse scenario could be observed, where DI into the primary isotopic envelope (red peak in Figure 2 right) occurred rapidly (within the first time point examined, 30 s), followed by a slower rate of incorporation in the secondary isotopic envelope (blue peak in Figure 2 right). Both the primary and secondary envelops eventually merged in the timeframe of the HDX experiment, as shown by the convergence of their *m*/*z* values. A number of peptides demonstrating this characteristic were detected in the HDX-MS data from rsHDL throughout the sequence of the apoA-I protein and are shown in the HDX kinetic plots in Appendix A and listed in Appendix A.

To establish the relationship between the observed HDX-MS data of apoA-I and its secondary structure in rsHDL, we compared the experimental HDX rate constants of backbone amide hydrogen atoms with theoretical rate constants for the unstructured (random coil) protein [57]. First, the HDX rate constants for peptides with the same sequence as the experimentally observed peptides were calculated for the conditions of the HDX reaction (i.e., pH = 6.8 (pD=7.2) and T = 0 °C), as described in the Materials and Methods section. For example, Figure 3 shows the DI MS envelops (Figure 3A) and HDX kinetic curves (Figure 3B) for an apoA-I peptide (residues T_161_–G_186_) that exhibited bimodal HDX kinetics, in which both fast (red) and slow (blue) exchanging peptides were observed (Figure 3A). The black dashed line (Figure 3B) represented the DI values predicted for a simulated unstructured peptide with the same sequence (T_161_–G_186_) as the apoA-I peptide by using HDX rate constants for individual residues. The two sets of experimental DI values (open triangles and circles) were fitted to stretched exponential functions that were drawn as colored curves (heat map defined in legend). The color indicated the average peptide HDX protection factor. The peptide relative abundance (Appendix A) was represented as a single or double solid line. The peptide represented by a double line was predicted to originate from an apoA-I dimer.

The experimentally derived HDX kinetic rate constants and the partitioning of peptide HDX sites using overlapping peptides (Materials and Methods) were used to probabilistically assign an HDX behavior to individual residues in each peptide. Subsequently, the HDX sites in peptides were aligned, and the HDX protection factors (Pf) were calculated for all residues (see Materials and Methods). Figure 3B shows at the top of the HDX kinetic graph a bar chart with the predicted Pfs of the residues in the T_161_–G_186_ peptide for each of the three apoA-I chains in rsHDL. Residues colored red/orange were fast H/D exchanged, while residues colored green/blue were slow H/D exchanged. The N-terminal residues of the digested peptides were exchanging very fast (faster than 30 s, the first experimentally recorded time point), and they were colored white because their HDX behavior could not be determined. The bar chart at the top of the graph and the peptide abundance data (Appendix A) indicated that the less abundant peptide corresponding to the T_161_–G_186_ domain (Appendix A) was located in the apoA-I hairpin chain and was predicted to undergo faster H/D exchange (i.e., many of its residues exchanged fast and were colored red/orange in the chart) relative to the more abundant peptide with the same sequence (Appendix A) located in the helical dimer and predicted to undergo slower H/D exchange. A significant number of residues in this peptide displayed slower H/D exchange and were colored blue in the chart. The proline residues did not have a backbone amide hydrogen and, therefore, were represented with black bars.

Note that in an earlier molecular dynamics simulation study of rHDL [25], we found that peptides with the same sequence from the Solar Flare regions of the apoA-I dimer [18,25] could have different conformations and subsequently distinct patterns of H/D exchange (bimodal HDX kinetics). This type of HDX kinetic behavior of apoA-I domains from the apoA-I dimer of rsHDL could be observed for peptides D_1_–T_16_, L_233_–Q_243_, and Y_236_–Q_243_, which had different conformations and/or experience different interaction environments (Appendix A).

### 3.4. HDX-MS Kinetic Analyses Support the Model of Reconstituted Spherical HDL with the Three ApoA-I Chains in a Dimer/Monomer Configuration

We extended the HDX kinetic analysis to all apoA-I digested peptides by noticing that many peptides that displayed bimodal HDX kinetics had, in general, relative abundances in the ratio 1:2, suggesting that peptides with relative intensity less than 1 (i.e., ~0.5) originated from an apoA-I single chain and not an apoA-I dimer. This observation was consistent with the helical dimer hairpin monomer (HdHp) model of rsHDL, wherein the three apoA-I polypeptide chains are configured as a combination of a single chain (hairpin monomer) and a helical dimer, with chains arranged anti-parallel [30]. In contrast, this observation was not consistent with two other alternative models of rsHDL (3Hp and iT3 [30]) in which the three apoA-I chains are in different configurations (e.g., three hairpin monomers (3Hp model) or an integrated trimer (iT3)). Subsequently, the HDX kinetic analysis for the entire set of peptides was performed on two separate peptide pools, one containing peptides with relative intensity one, and the other containing peptides with a relative intensity <1. Peptides that do not exhibit bimodal HDX kinetics were added to both pools. The relative intensities of the peptides displaying bimodal HDX kinetics were estimated from their deconvoluted isotopic envelopes recorded at the first experimental time point (30 s) when their centroid masses were at the largest separation (Figure 3A). The relative intensity of the slower HDX and more abundant peptide (blue cartoon) in Figure 3A was taken as one, while the ratio of the cartoon heights (red to blue) was taken as the relative intensity of faster HDX and less abundant peptide (red cartoon). This ratio was ~0.5 for peptide T_161_–G_186_ (Figure 3A, Appendix A).

Figure 4A and Appendix A show all apoA-I peptides of rsHDL detected by MS. The average Pfs for all peptides were calculated and color-coded. Each bar in the map (Figure 4A) represented one digested peptide if it was unicolor, or two bimodal HDX peptides if it was bicolor. The thickness of the color strips in bicolor bars was proportional to their relative abundance. Peptides colored red had small protection factors (Pf < 10) and exchanged their backbone amide hydrogen almost as fast as peptides in random coil conformation. In contrast, those peptides that were colored blue displayed the slowest H/D exchange (Pf > 10^3^).

We noticed that the abundances of digested peptides with the same sequence displaying bimodal HDX could be either the same or in a ratio of 2:1. This would suggest that bimodal HDX peptides with abundances in a 2:1 ratio might originate from a monomer-dimer configuration of the three apoA-I chains. Based on this observation, we classified all peptides into two groups, corresponding to a dimer or a monomer, and built overlapping peptide maps separately for the monomer and dimer (Figure 4B). The map for peptides predicted to originate from an apoA-I dimer is shown at the top of Figure 4B, and for the monomer at the bottom. Because all bimodal HDX peptides were assigned either to the dimer or the monomer based on their abundance, all bars in these two maps were unicolor. By comparing the average Pfs of peptides in the dimer’s map vs. the monomer’s map, we could conclude that the apoA-I monomer in rsHDL was significantly more dynamic than the dimer because the former had faster-exchanging HDX peptides (red) than the latter. We also noted that the monomer was more dynamic than the dimer in the N-terminus domain, and the residue ranged 110–160 of apoA-I but had a less dynamic C-terminus.

The LCAT binding domain on apoA-I (the so-called Solar Flare loop, L_159_–A_180_) in rsHDL was predicted to be as dynamic as in rHDL [18], but it exchanged faster when located in the hairpin monomer than in the helical dimer. The apoA-I domain E_125_–A_158_ was reported to exhibit bimodal HDX kinetics in rHDL [56] and sHDL isolated from plasma [31]. We looked to see whether there was a distinction in HDX behavior among the three apoA-I chains at this region of rsHDL. As with the LCAT binding domain, we predicted that, when located in the hairpin monomer, this apoA-I domain (E_125_–A_158_) was more dynamic than when located in the helical dimer.

### 3.5. HDX Bimodal Kinetics Reveals a Complex Conformational Dynamics of ApoA-I in Reconstituted Spherical HDL

Measurements of rsHDL samples by small-angle neutron scattering (SANS) with contrast variation suggested that the three apoA-I chains could be arranged in three distinct configurations to fit the SANS low-resolution structure: the helical dimer hairpin monomer (HdHp) model, the integrated trimer (iT3) model, and the three hairpin monomers (3Hp) model [30]. When MS cross-linking data were considered too, the most likely organization of the three apoA-I chains seemed to be a combination of the monomeric hairpin and the helical dimer (i.e., the HdHp model) [30]. In addition, we showed previously that there were apoA-I domains in rHDL (reconstituted with 1,2-dimyristoyl-*sn*-glycero-3-phosphocholine, rHDL_DMPC_) that exhibited bimodal HDX kinetics [37], and that both fast and slow H/D exchange behavior of protein domains with the same amino acid composition originated in alternative conformations and/or distinct molecular environments. For example, we showed that the apoA-I domain Y_115_–L_126_ in rHDL_DMPC_ experienced two distinct conformations and environments, corresponding to the locations of this domain either in the apoA-I hairpin domain, where it is predicted to have reduced interaction with lamellar phase lipids, relative to outside the hairpin domain, where it is predicted to be directly interacting with the lamellar lipid phase [37].

A similar scenario seemed to apply to rsHDL, as well. For example, the digested peptide E_128_–M_148_ exhibited bimodal HDX kinetics (Figure 5A). Peptides corresponding to the apoA-I region that underwent faster HDX (open circles, red line) were half as abundant as peptides allegedly produced by the apoA-I dimer, undergoing slower HDX (open triangle, double orange line). If the HdHp model of rsHDL [30] was assumed, then it was conceivable that E_128_–M_148_ had two distinct conformations, depending on which apoA-I chain was located, as shown in Figure 5B. E_128_–M_148_ was more dynamic when located on the apoA-I hairpin monomer (Pf = 1, red) than when located in the helical dimer (Pf = 14, orange). The bar chart at the top of the HDX kinetic graph in Figure 5A predicted that most of E_128_–M_148_ residues, when located in the hairpin monomer, had Pf < 10 (colored red), while when located in the helical dimer, one-third of residues had Pf > 100 (colored green), corresponding to slower HDX.

In a previous publication [18], we reported that the apoA-I domain L_159_–A_180_, the binding site for LCAT, the enzyme that remodels nascent HDL into spherical HDL, was highly dynamic. In rsHDL, we found that peptides within the L_159_–A_180_ domain (e.g., L_159_–L_170_ and R_171_–E_179_) exhibited bimodal HDX kinetics (Appendix A).

In both HDX modes of L_159_–L_170_, six/seven residues out of eight exchanged very fast (Pf < 10, Appendix A), as illustrated by the single (open circle) and double (open triangle) red curves in Figure 6A. In contrast, the two HDX modes of R_171_–E_179_ were exchanging slower than those of L_159_–L_170_ (two or four residues out of six exchanged slowly, Pf > 2000, Appendix A). The location of the domain L_159_–L_170_ in the apoA-I chains of rsHDL (the HdHp model of [30]) is shown in Figure 6B. The same domain located in the helical dimer was as dynamic as when it was located in the hairpin monomer, and our direct experimentally observed HDX kinetic behavior of the L_159_–A_180_ region of apoA-I in rsHDL confirmed the intrinsic dynamic nature of this domain of apoA-I regardless of whether the HDL particles contained two or three molecules of apoA-I. This was consistent with our previous finding for this region of apoA-I in rHDL [18].

Different combinations of non-overlapping digested peptides (Figure 4) could be used to create probable conformations of the three apoA-I chains in rsHDL. As an example (Figure 7), we used the first row of peptides from Figure 4B to construct conformations separately for the apoA-I dimer and the hairpin monomer and mapped the selected peptides to the 3-D models (HdHp, iT3, 3Hp) of rsHDL proposed previously based on SANS [30]. The conformations obtained with this particular set of peptides were overall very dynamic, displaying relatively low protection factors to D incorporation (Pf < 100) for most of the protein chain. The hairpin monomer was more dynamic than the dimer, which was indicated by the presence of larger, very dynamic peptide domains (Pf < 10) in the monomer than in the dimer. We also found interesting that a very dynamic (Pf < 10) N-terminus of one apoA-I chain in the dimer coupled to a much less dynamic (Pf > 100) C-terminus of the other chain oriented antiparallel. The pattern of DI along apoA-I chains obtained from the HDX data could not discriminate among the three models, as these data represented local dynamics behavior and were not related with the overall shape, but the abundances of deuterated peptides (Appendix A) suggested, as the SANS data [30], that the HdHp model was most likely to represent the actual configuration of the three apoA-I chains in rsHDL.

Only one other HDX study performed on spherical HDL isolated from human plasma was previously reported by Chetty et al. [31]. However, the sHDL preparations used in this study were not monodisperse. They contained mostly sHDL particles with four, five, and six apoA-I molecules per particle (Figure 2 in [31]) and varied in size from 8.1 to 10 nm (Figure 1 in [31]). Because our preparations of reconstituted sHDL contained three apoA-I molecules per particle and were 8.8 nm in diameter, it was difficult to compare the two studies in terms of HDX rate constants and protection factors. Nevertheless, there are similarities and differences between the two studies that are worth discussing. For example, Chetty et al. [31] reported slower HDX kinetics for plasma sHDL than we observed on rsHDL. This discrepancy could be attributed to the differences in particle composition and was supported by our EINS measurements in which an HDL particle with two apoA-I molecules (rHDL) was shown to be more dynamic than one with three apoA-I molecules. The higher protein content of the plasma sHDL particles used by Chetty et al., compared to our samples, might also explain why we observed a richer dynamic range for apoA-I chains with many digested peptides exhibiting bimodal HDX kinetics. However, both HDX studies agreed that the region 115–158 of apoA-I exhibited bimodal HDX kinetics (Figure 4 and Figure 5). Finally, Chetty et al. concluded that their HDX analysis of plasma sHDL with 4–6 apoA-I molecules per particle was consistent with apoA-I being configured in a trefoil type arrangement [29], while we concluded that the three apoA-I molecules in rsHDL were configured in a dimer-monomer arrangement.

While the HDX measurements were performed at a single temperature (273.15 K) and gauged local dynamics of the protein backbone and conformational diversity only, the EINS measurements gauged the overall softness and flexibility (dynamics) of the particle as a whole (protein and lipids) through the mobility of the H atoms from lipids and protein alike. The two methods did not corroborate each other’s findings but were complementary in the sense that they revealed different information about the particle dynamics. Thus, it was not possible to infer from HDX data the force constant for the overall dynamics of the entire particle (protein and lipid) obtained from the EINS measurements, or the HDX rate constants from EINS measurements. Nevertheless, both EINS and HDX methods are well-established techniques to explore molecular dynamics, but different facets of that dynamics. EINS can be applied to any macromolecular assembly, while HDX is limited to proteins because the backbone amide H experiences a slow exchange with D, which can be captured within the time frame of the experiment.

## 4. Conclusions

We performed elastic incoherent neutron scattering (EINS) measurements and hydrogen-deuterium exchange mass spectrometry (HDX-MS) kinetic analyses of physiologically competent rHDL and rsHDL particles reconstituted with human apoA-I, POPC, unesterified cholesterol, and cholesteryl esters to determine the dynamics of the protein backbone (local dynamics) and the overall particle softness and flexibility (overall dynamics). HDLs have the highest protein-to-lipid ratio and are the smallest and densest of all lipoprotein classes. Pure lipids are highly dynamic at ambient temperatures [4,66,70,71], and one would expect that the mobility of lipoproteins to increase with the increase in their lipid content. For example, rHDL has ~26% protein mass and ~74% lipid mass, whereas rsHDL has ~37% protein mass and ~63% lipid mass, so rHDL should be softer and more flexible than rsHDL. According to our neutron scattering (EINS) results, rHDL was indeed softer and more flexible than rsHDL at T > 250 K. The mean square displacements calculated from EINS measurements in the temperature range 20–310 K showed three distinct domains of particle softness that could be associated with distinct force constants for particle mobility. However, studies on natural membranes from hyperthermophilic and mesophilic bacteria revealed that the protein-to-lipid ratio might not be the only parameter that determines the dynamic behavior of a biological particle that contains both lipids and proteins [72].

The comparison of rHDL and rsHDL dynamics, as determined by EINS (rHDL vs. rsHDL in Figure 1) with two other plasma lipoproteins previously studied using EINS:VLDL vs. LDL (Figure 1 in [5]), revealed that VLDL and rHDL (the bigger lipoprotein particles in the two pairs) were softer and more flexible than LDL and rsHDL at ambient temperature, respectively. During VLDL maturation into LDL, the more dynamic triglycerides were removed, and the more rigid cholesteryl esters were incorporated, leading to apoB100 protein rearrangement and particle shrinking (e.g., from ~50 nm to ~20 nm [73,74]). Similarly, rsHDL particle was smaller and more compact than rHDL (8.3 vs. 9.6 nm, respectively), was rich in cholesteryl esters, and had more protein content (apoA-I or apoA-II). These structural changes in VLDL and rHDL (when remodeled into LDL and rsHDL, respectively) seemed to be mirrored by changes in particle dynamics. Various plasma enzymes (e.g., LCAT, PLTP, CETP) and cell receptors (e.g., SR-BI) changed HDL dynamics as it matured from nascent (discoidal) to a spherical form. It was conceivable that the change in the softness and flexibility of HDL particles during their life span contributed to the selective interaction of the HDL protein components (apoA-I, apoA-II) with these enzymes, which are critical for HDL remodeling and performance of its main physiological function of carrying lipid cargo in reverse cholesterol transport. For example, lecithin cholesteryl acyltransferase (LCAT), which esterifies the free cholesterol on the surface of HDL, seemed to easily attach to a softer and more flexible discoidal HDL particle by interacting with specific and more flexible domains of apoA-I (e.g., the Solar Flare loop). In contrast, other plasma enzymes, like the phospholipid (PLTP) and cholesteryl ester (CETP), transfer proteins appeared to more preferentially interact with a more rigid HDL particle (spherical HDL) in order to accomplish their lipid transfer functions. Thus, the HDL became less dynamic only after cholesterol was converted to cholesteryl ester, and the biophysical change, in turn, impacted its interaction with remodeling partners.

Overall, our HDX kinetic analysis of apoA-I in rsHDL indicated that bimodal HDX kinetics, observed in many of the apoA-I peptides, came from conformational diversity of apoA-I chains and/or the presence of distinct interaction environments. The distinct dynamic behavior of apoA-I chains in rsHDL suggested that they were configured as dimer/monomer, consistent with the HdHp model of rsHDL. Lastly, our HDX kinetics analysis revealed that the Solar Flare domain of apoA-I (L_159_–A_180_), the LCAT binding site, earlier proposed to exist in nearly random coil conformation in rHDL [18], displayed bimodal HDX kinetics in rsHDL and was in nearly random coil conformation.

## Figures and Tables

**Figure 1 biomolecules-10-00121-f001:**
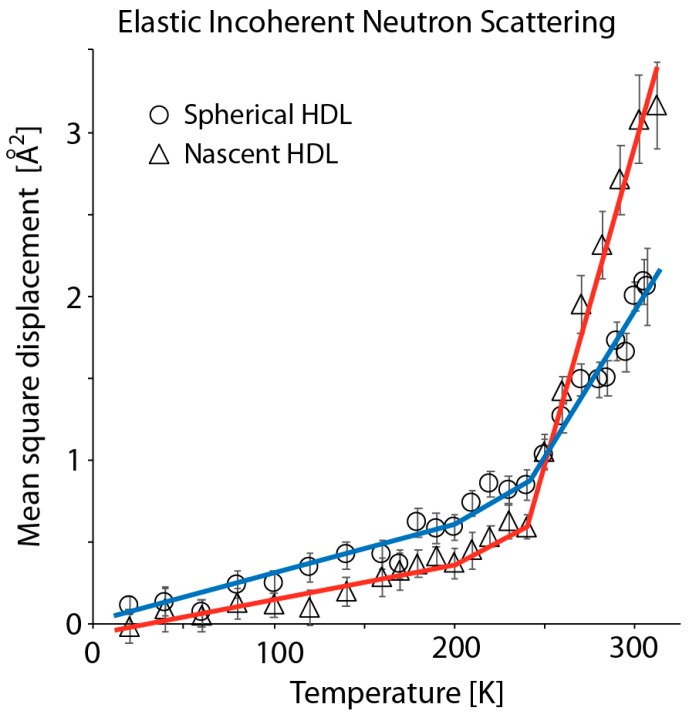
Mean square displacement (MSD) of atoms in reconstituted discoidal and spherical high-density lipoprotein (HDL) (hydrated powder) calculated from elastic incoherent neutron scattering (EINS) data collected on spectrometer IN13. The graph shows a comparison of MSD obtained from measurements of reconstituted spherical HDL particles (rsHDL) (open circle) and reconstituted discoidal HDL (rHDL) (open triangle) in the temperature range 20–310 K. The variation of MSD with temperature showed three linear domains that could be related with three different force constants for particle dynamics. The linear domains were marked with line segments (blue for rsHDL, and red for rHDL). The two MSD curves intersected at ~250 K. Overall, rsHDL was slightly more dynamic than rHDL for T < 250 K, but the latter became much more dynamic at ambient temperature.

**Figure 2 biomolecules-10-00121-f002:**
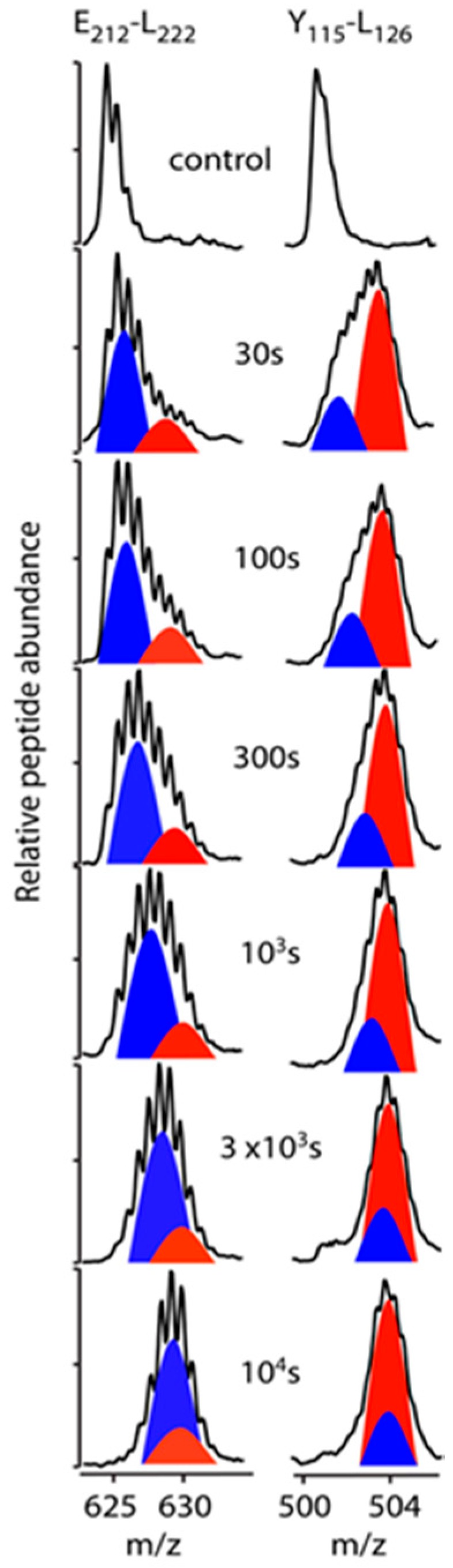
Bimodal hydrogen-deuterium exchange (HDX) kinetics in apolipoprotein A-I (apoA-I) in rsHDL. The change in the shape of the MS peak of a digested peptide as a function of the HDX reaction time was an indication that the protein domain, corresponding to the digested peptide, existed in different conformations or experienced different interaction environments in the folded protein. *Left*: The MS spectra of peptide E_212_DLRQGLLPVL_222_ at various HDX times suggested that the corresponding protein domain in the intact protein existed in two distinct conformations, one in which the amide H was better protected from HDX, for example, in an alpha-helical arrangement, leading to a slow H to D exchange (blue peak), and a second conformation in which H was replaced by D much faster (red peak). The difference in HDX rates between the two domains led to a characteristic shape for the MS peak, a shape that also depends on the abundance of the two conformations. In this case, the fast exchanging protein domain produced a peptide abundance (red peak) about half of the slow H/D exchanging domain. *Right*: Bimodal HDX kinetics is shown for the case of a slow H/D exchanging peptide (blue peak, Y_115_RQKVEPLRAEL_126_) that had a peptide abundance about half of the fast H/D exchanging peptide (red peak). The MS spectra for each peptide and the peptide peaks showed that the centroids (*m*/*z*) for each “peak” (red/blue cartoons) were gradually merging at longer HDX times (10^4^ s).

**Figure 3 biomolecules-10-00121-f003:**
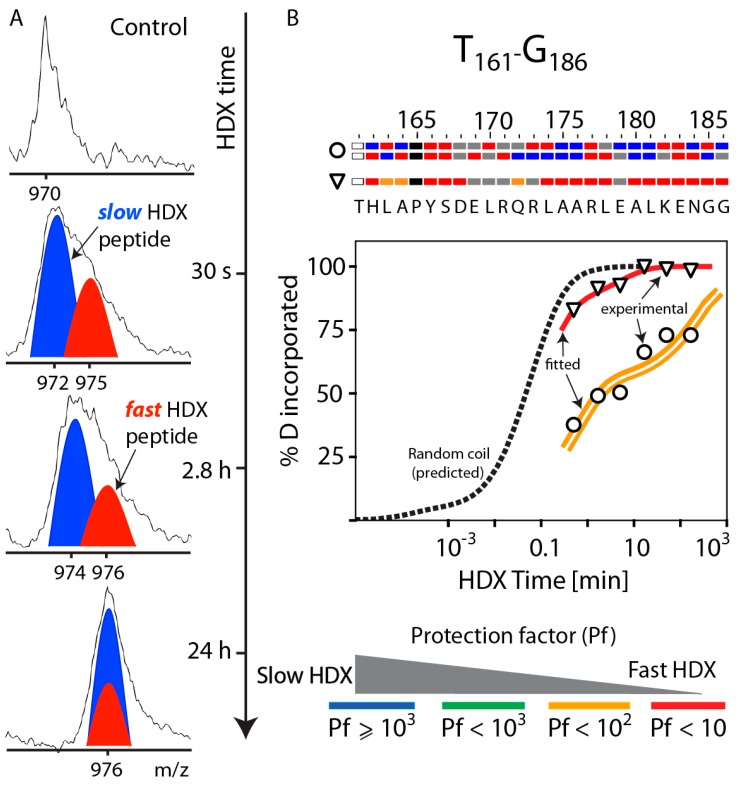
Bimodal and biphasic HDX kinetics of apoA-I peptide T_161_–G_186_ in rsHDL. (**A**), MS spectra at various HDX times were “partitioned” into “spectra” (colored cartoons) for slow (blue) and fast (red) H/D exchanging protein domains. The peak centroids for the two protein domains merged over time, as the slower HDX domain became fully deuterated. The abundance of the peptide corresponding to the slower HDX domain was estimated to be about twice that of the peptide corresponding to the faster HDX domain. (**B**), Kinetic analysis of the HDX-MS spectra: the dotted black line displayed D incorporation in the T_161_–G_186_ domain of apoA-I if in random coil conformation. The experimental D incorporation for T_161_–G_186_ were shown with open circles (slow HDX) and open triangles (fast HDX), and the stretched exponential functions fitted to the experimental values (open circles and triangles) were the red and orange curves, respectively. The fact that the slow exchanging domain produces peptides twice as abundant as those generated by the fast exchanging domain was indicated with a double line (orange) and was consistent with the idea suggested earlier that two of the three apoA-I polypeptide chains (T_161_–G_186_ region) in rsHDL experience an interaction environment that leads to slow HDX. T_161_–G_186_ exhibited both bimodal and biphasic HDX kinetics. A protein domain exhibited biphasic HDX kinetics when the amino acid residues within the domain exchanged H/D with different HDX rates. The HDX rate constants for the slow and fast HDX domains could be obtained by fitting the DI values to stretched exponential functions. The HDX protection factors (Pf) for amino acid residues in the T_161_–G_186_ domain (in each apoA-I molecule) were shown color-coded in the bar chart at the top of the graph. The color-coding scheme is shown at the bottom and is as follows: red: Pf < 10, orange: 10 ≤ Pf < 100, green: 100 ≤ Pf < 1000, blue: Pf ≥ 1000; residues with unknown HDX behavior were colored grey. The T_161_–G_186_ domain originating from the apoA-I dimer was predicted to exchange slower than the one located in the hairpin chain, possibly because apoA-I dimerization reduced the dynamics of its monomers.

**Figure 4 biomolecules-10-00121-f004:**
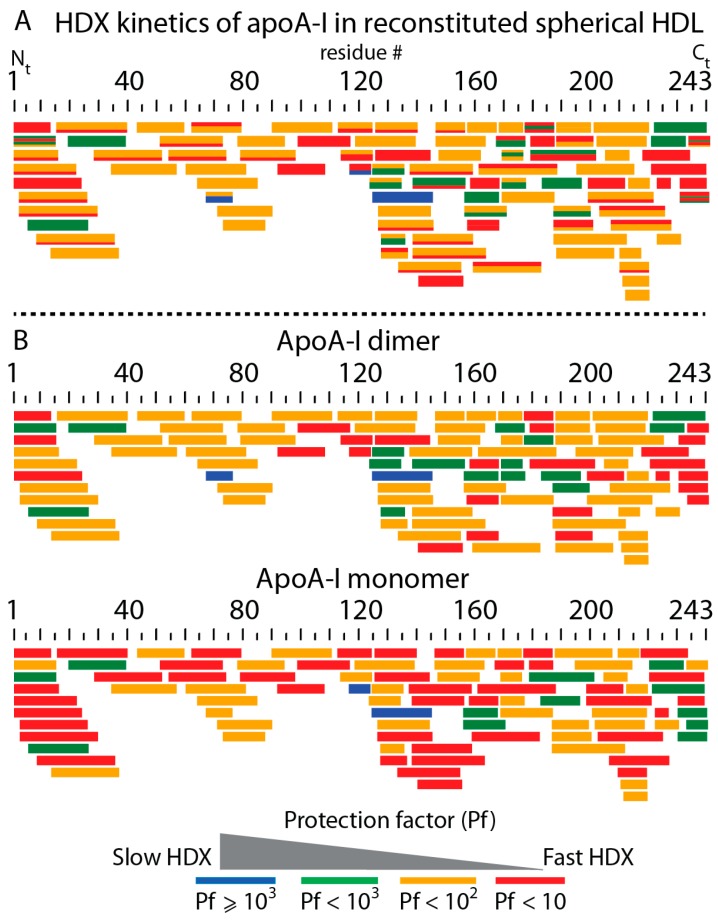
Map of all overlapping digested apoA-I peptides observed in the HDX-MS experiments. (**A**) Peptides were depicted as colored bars; bicolor bars represent peptides with bimodal HDX kinetic behavior. The thickness of colored stripes in a bar was proportional to the peptide’s relative abundance. (**B**) Digested peptides, displaying bimodal HDX kinetics, were grouped in two pools: the top bar chart shows the set of digested peptides predicted to originate from an apoA-I dimer. The peptides were color-coded according to their average HDX protection factor (Pf). The bottom bar chart depicts peptides predicted to originate from an apoA-I hairpin monomer. A comparison of the top and bottom charts revealed that the apoA-I hairpin monomer in rsHDL was more dynamic than the dimer.

**Figure 5 biomolecules-10-00121-f005:**
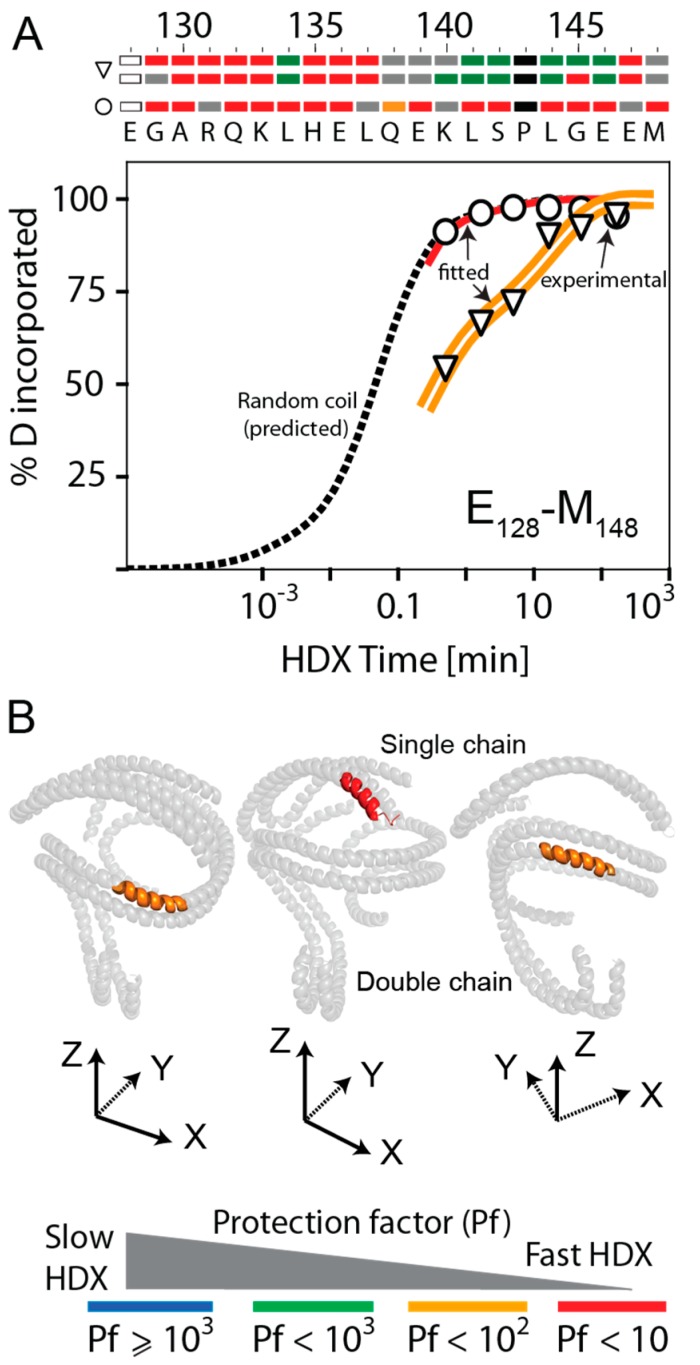
Kinetic curves and HDX residue protection factors for peptide E_128_–M_148_. The color-coding legend of the curves is shown at the bottom of the figure. (**A**) Peptide E_128_–M_148_ exhibited bimodal and biphasic HDX kinetics (red and orange curves). The faster-exchanging peptide (red) behaved like a random coil peptide and was predicted to originate from the apoA-I hairpin, while the slower exchanging peptide was predicted to come from the helical double chain (orange double line). (**B**) The apoA-I domain E_128_–M_148_ identified to exhibit bimodal HDX kinetics in rHDL, as shown mapped on the three apoA-I chains of rsHDL (HpHd model). The domain was predicted to exchange fast when located in the hairpin monomer (the majority of residues were colored red) but was exchanging slower when located on the apoA-I helical dimer (many of its residues were colored green).

**Figure 6 biomolecules-10-00121-f006:**
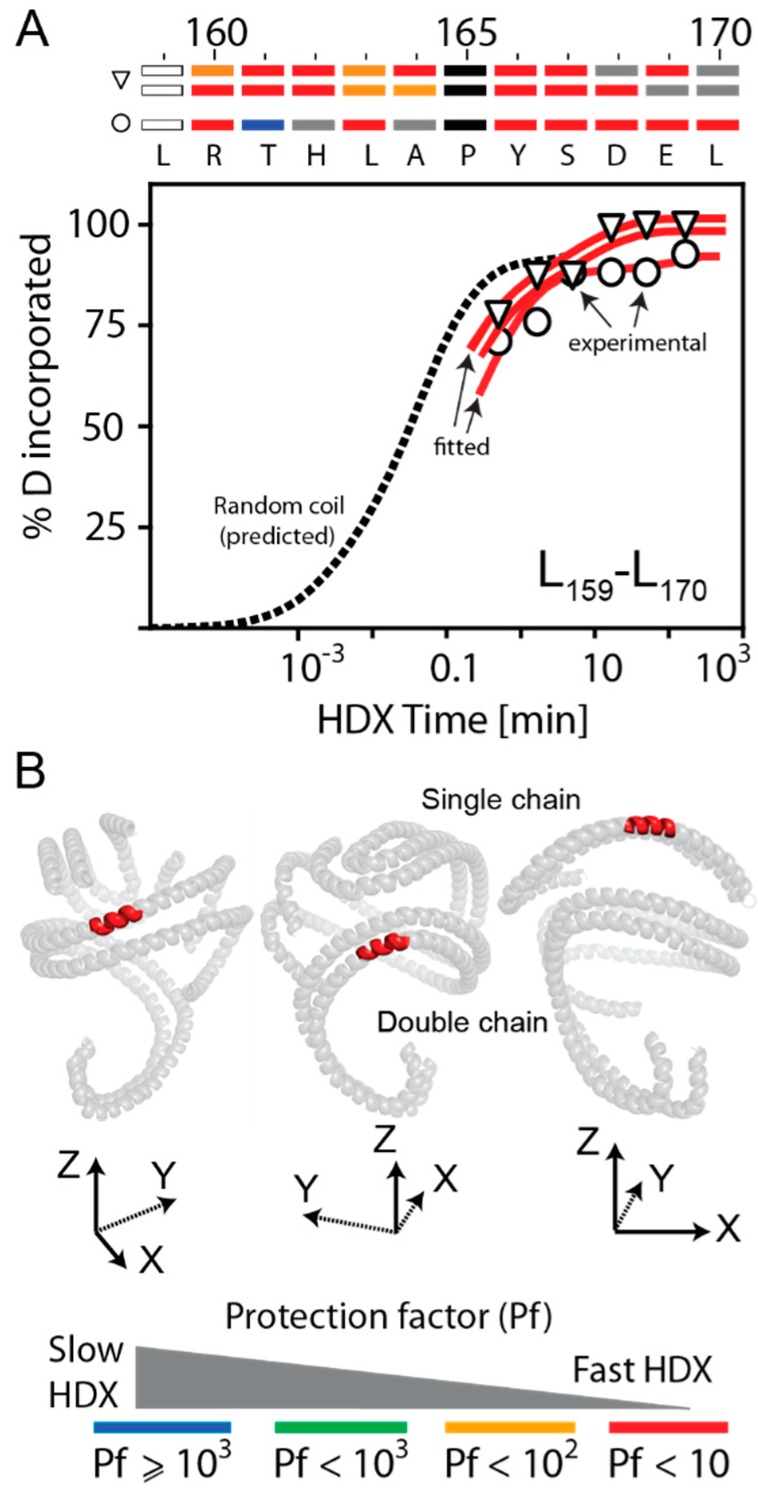
Kinetic curves and HDX residue protection factors for peptide L_159_–L_170_ of the Solar Flare region of apoA-I (LCAT binding site) in rsHDL. The experimental DI values were shown with open circles and triangles. (**A**) Time-dependent HDX behavior of peptide L_159_–L_170_. This peptide exhibited bimodal and biphasic HDX kinetics, and both exchange modes (single and double red curves) were rather fast (Appendix A). The protection factors (Pf) of the residues in L_159_–L_170_ for all three chains were shown in the bar chart at the top of the graph; residues with undetermined HDX behavior were colored grey. Most of the residues in this peptide, corresponding to half of the Solar Flare region of apoA-I, were exchanging fast (colored red in the bar map shown on top of the HDX kinetic graph). (**B**) Location of the apoA-I domain L_159_–L_170_ in the three apoA-I molecules of rsHDL arranged in the monomer/dimer configuration predicted by the HdHp model. The HDX data showed that this protein domain was rather dynamic (Pf_avg_ < 10) for all apoA-I molecules.

**Figure 7 biomolecules-10-00121-f007:**
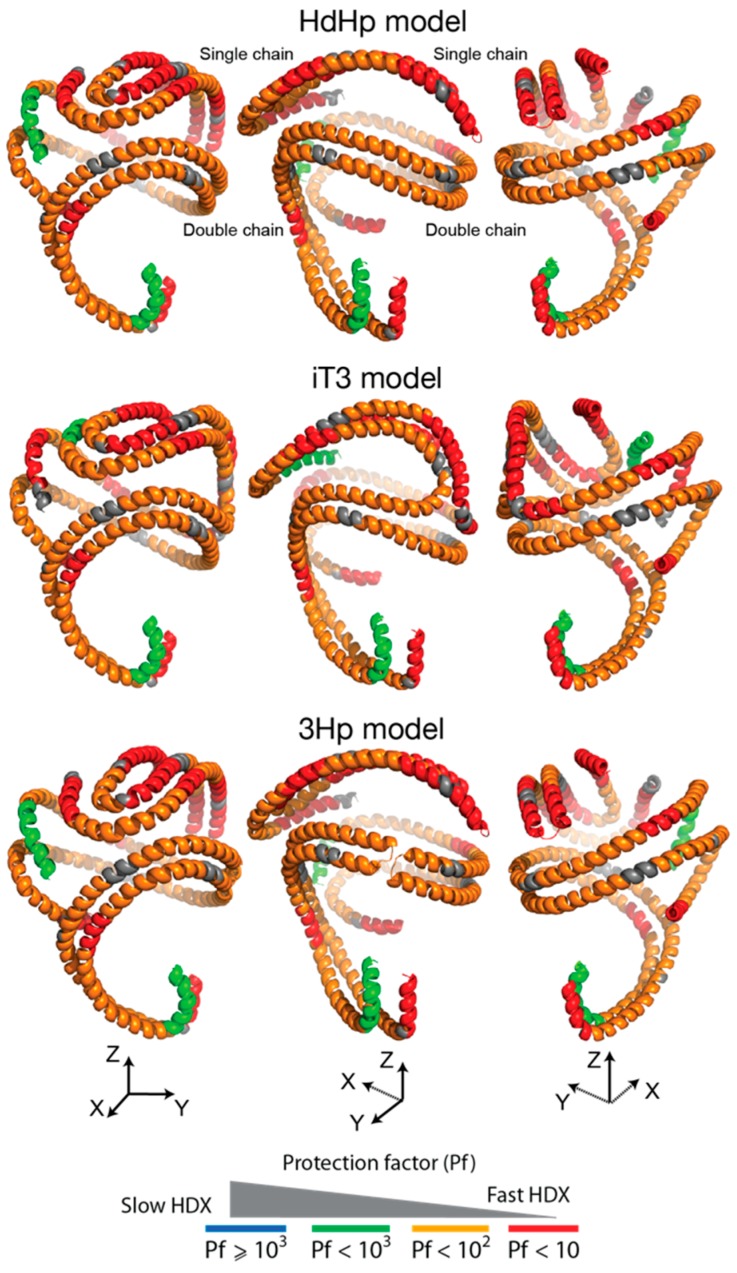
The pattern of HDX protection factors for apoA-I domains in rsHDL (corresponding to digested peptides) mapped on rsHDL models obtained from small-angle neutron scattering: the helical dimer hairpin monomer (HdHp) model (top), the integrated trimer (iT3) model (middle), and the three hairpin monomers (3Hp) model (bottom). The 3D HDX maps show that overall the apoA-I hairpin monomer was more dynamic than the apoA-I helical dimer.

**Table 1 biomolecules-10-00121-t001:** Effective force constants for particle dynamics calculated from average mean square displacement for various temperature ranges.

**Lipoprotein**	**Instrument *^a^***	**Force Constant *^b^***		
		20–200 K ***^c^***	200–250 K	250–310 K
rsHDL ***^d^***	IN13	1.00 ± 0.13	0.25 ± 0.03	0.15 ± 0.03
rHDL ***^e^***		1.13 ± 0.11	0.49 ± 0.11	0.067 ± 0.002
rsHDL	IN16	0.88 ± 0.03	0.17 ± 0.01	0.12 ± 0.02

***^a^*** The spectrometer used to measure the elastic incoherent neutron scattering, ***^b^*** Average force constant [N/m] of the molecular motion calculated with Equation (3) (see Methods), ***^c^*** Range of absolute temperature [K], ***^d^*** reconstituted spherical HDL, ***^e^*** reconstituted nascent discoidal HDL.

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
