# Peer review of "Protein Backbone and Average Particle Dynamics in Reconstituted Discoidal and Spherical HDL Probed by Hydrogen Deuterium Exchange and Elastic Incoherent Neutron Scattering"

_biomolecules, 2020, doi:10.3390/biom10010121_

Round 1

Reviewer 1 Report

The authors analyzed structure of the reconstituted discoid HDL with apoA-I and POPC and cholesterol, as well as the spherical particles subsequently produced remodeling this precursor by the LCAT reaction to create a core lipid molecule of esterified cholesterol, by using technologies of Hydrogen deuterium exchange and elastic incoherent neutron scanning. They concluded that the particles thus produced contain average 3 apoA-I molecules. Their analysis indicated molecular organization of these three apoA-I molecules as one hairpin-shaped monomer and two antiparallel-oriented molecules. Their model seems solid and reasonable as far as the structure of this model particles. The question is how these models can be extended to the real world of HDL particles in blood plasma.

Critiques

The most simple and classical model for plasma HDL was provided by Shen, Scanu and Kezdy based on chemical composition of the particles and spherical microemulsion model (PNAS 74: 837-841, 1977). It is wondered how the current data fit or would be consistent with this model structure since it was based on very simple geometric analysis of the particles and yet interpret general structure of lipoproteins. More recent bioinformatics analysis of human plasma HDL indicated that apoAII-containing HDL has the structural and metabolic natures distinct from apoA-I-only HDL, but both seem based on the particles containing 3 to 4 apoA-I molecules and apoA-II containing HDL is merely a product of statistical distribution of both apolipoproteins (Sci Rep 6: 31532, 2016). This means that the current model by the authors may represent this prototype HDL particles. On the other hand, any helical apolipoprotein can generate discoidal nascent HDL in the presence of ABCA1 (JBC 266: 3080-3086, 1991). How the authors may view generation of apoA-II-containing HDL from this prototype and its metabolic fate?

Author Response

We thank the reviewer for the positive and constructive critiques that helped us improve the manuscript significantly. Bellow we answer reviewers’ critiques one by on:

Reviewer’s comment:

The authors analyzed structure of the reconstituted discoid HDL with apoA-I and POPC and cholesterol, as well as the spherical particles subsequently produced remodeling this precursor by the LCAT reaction to create a core lipid molecule of esterified cholesterol, by using technologies of Hydrogen deuterium exchange and elastic incoherent neutron scanning. They concluded that the particles thus produced contain average 3 apoA-I molecules. Their analysis indicated molecular organization of these three apoA-I molecules as one hairpin-shaped monomer and two antiparallel-oriented molecules. Their model seems solid and reasonable as far as the structure of this model particles. The question is how these models can be extended to the real world of HDL particles in blood plasma.

Response to reviewer’s comment:

We have been using reconstituted nascent (2 apoA-I) and spherical (3 apoA-I) HDL preparations for many years, and others and we have shown in previous publications that reconstituted HDL, while less complex in lipid composition, have a similar protein lipid composition with the nascent and spherical HDL particles generated within the vascular compartment. Moreover, we have shown that these particles are biologically active, promoting cholesterol efflux, interacting with HDL receptor (SRB1), and interacting with lipid transporters like ABCA1, and ABCG1. These particles also interact with multiple HDL remodeling enzymes like LCAT, ACAT, PLTP and CETP. The structurally characterized reconstituted HDL possess numerous HDL related biological activities.

We also would like to respectfully point out that the present study does not proposes new models for the structure of HDL, but rather focuses on investigating, for the first time, dynamics characteristics of HDL, which are as relevant to its physiological functions as its structure is. We believe that our conclusions on the dynamics of reconstituted HDL are applicable to plasma HDL because of the similarity in their protein lipid composition, and the ability of the reconstituted particles to function biochemically with receptors and lipid remodeling transporters comparably to plasma HDL particles.

Finally, the reviewer’s question is well taken – but we believe it can be raised for any crystal structure, or alternative recombinant reconstituted macromolecular structure.  Demonstration that multiple biophysical and biological properties are comparable and shared, serves as a cornerstone for the field in accepting reconstituted HDL particles as a reasonable facsimile for HDL particles in vivo.

Reviewer’s critique:

The most simple and classical model for plasma HDL was provided by Shen, Scanu and Kezdy based on chemical composition of the particles and spherical microemulsion model (PNAS 74: 837-841, 1977). It is wondered how the current data fit or would be consistent with this model structure since it was based on very simple geometric analysis of the particles and yet interpret general structure of lipoproteins. More recent bioinformatics analysis of human plasma HDL indicated that apoAII-containing HDL has the structural and metabolic natures distinct from apoA-I-only HDL, but both seem based on the particles containing 3 to 4 apoA-I molecules and apoA-II containing HDL is merely a product of statistical distribution of both apolipoproteins (Sci Rep 6: 31532, 2016). This means that the current model by the authors may represent this prototype HDL particles. On the other hand, any helical apolipoprotein can generate discoidal nascent HDL in the presence of ABCA1 (JBC 266: 3080-3086, 1991). How the authors may view generation of apoA-II-containing HDL from this prototype and its metabolic fate? 

Response to reviewer’s critique:

We agree with the reviewer that early studies on plasma HDL provided a basic understanding of HDL structure and composition. However, the HDL structure field has advanced considerably since the papers mentioned by the reviewer were published and include the use of novel techniques such as small angle X-ray and neutron scattering.

We also agree with the reviewer that investigating the structure and the dynamics of HDL particles reconstituted with both apoA-I and apoA-II should be very interesting as both type of particles exist in plasma, however such an endeavor is beyond the scope of our present study and in our experience reconstituting HDL with both apoA-I and apoA-II turned out to be challenging. We note also that most HDL particles in plasma do not possess an apoA-II, with only a minority having this smaller and less abundant apolipoprotein. Nevertheless, we hypothesize that while the structure of an HDL particle containing both apoA-I and apoA-II might be different from the point of view of protein configuration, the dynamic characteristics, as gauged by EINS, of HDL particles containing either apoA-I/apoA-II or apoA-I only should be similar if their protein/lipid ratio is the same. The difference in lipoprotein remodeling between HDL particles with/without apoA-II, as discussed in Sci Rep 6: 31532, 2016, may come from a difference in the interaction with plasma enzymes and cell receptors due to presence of apoA-II.

We added a paragraph in the Results/Discussion section about HDL particles containing apoA-I/apoA-II (Lines: 420-426):

“A recent study reported that plasma HDL containing apoA-I/apoA-II participates less in lipoprotein remodeling than HDL containing apoA-I only, and that there is little difference between the metabolic fate of HDL particles with or without apoA-II (Kido et al., Sci Rep 6: 3153). We hypothesize that the dynamics of HDL particles containing either apoA-I or apoA-I/apoA-II should be similar if their protein ratio is the same. The difference in the ease with which these distinct lipoproteins remodel likely stems from changes in interaction with plasma enzymes and cell receptors due to the presence of apoA-II, rather than differences in overall dynamic behavior.”

Reviewer 2 Report

In the present study, the authors applied different spectroscopic techniques to evaluate apoA-I structure and lipoprotein flexibility in reconstituted spherical HDL. Overall, the application of EINS and HDX-MS confirmed previous findings from the same authors and did not help to clarify whether the HdHp, iT3 or 3Hp hairpin model is likely the structure of three apoA-I molecules within a spherical HDL particle.

Although very detailed from a technical point of view, it is difficult to understand the key messages and their biological impact.

For EINS studies, reconstituted particles were lyophilized and rehydrated at 200 mg/ml. The authors stated that this procedure did not affect particle morphology and homogeneity: data supporting this statement must be provided.

Author Response

We thank the reviewer for the positive and constructive critiques that helped us improve the manuscript significantly. Bellow we answer reviewer’s critiques one by on:

Reviewer’s critique:

In the present study, the authors applied different spectroscopic techniques to evaluate apoA-I structure and lipoprotein flexibility in reconstituted spherical HDL. Overall, the application of EINS and HDX-MS confirmed previous findings from the same authors and did not help to clarify whether the HdHp, iT3 or 3Hp hairpin model is likely the structure of three apoA-I molecules within a spherical HDL particle.

Response to reviewer’s critique:

We would like to emphasize that our hydrogen deuterium exchange (HDX) data on spherical HDL not only supports a previously published small angle neutron scattering (SANS) model of spherical HDL (the HdHp model), but also reveals a very intricate dynamics of the three apoA-I molecules within the HDL particle, which was not reported before. We respectfully disagree with the reviewer’s assessment that the HDX data was not helpful in discriminating between the 3 models of spherical HDL proposed by Wu et al. (JBC 2011). Actually the fact that the abundances of many digested peptides are in a ratio 2:1 strengthens the conclusion that the 3 apoA-I molecules are organized in a dimer : monomer configuration (i.e. that two apoA-I experience similar environments relative to a third apoA-I). The same conclusion is supported by the HDX pattern for individual digested peptides shown in Fig. 4. In the Results and Discussion sub-section: HDX-MS kinetic analyses support the model of reconstituted spherical HDL with the three apoA-I chains in a dimer/monomer configuration we state:

“…many peptides that display bimodal HDX kinetics have, in general, relative abundances in the ratio 1:2 suggesting that peptides with relative intensity less than 1 (i.e. ~0.5) originate from an apoA-I single chain and not an apoA-I dimer. This observation is consistent with the HdHp model of rsHDL wherein the three apoA-I polypeptide chains are configured as a combination of single chain (hairpin monomer) and a helical dimer with chains arranged anti-parallel. In contrast, this observation is not consistent with two other alternative models of rsHDL (3Hp and iT3 Wu et al. JBC 2011) in which the three apoA-I chains are in different configurations (e.g. three hairpin monomers (3Hp model) or an integrated trimer (iT3)).”

In the Conclusion section we state:

“Overall, our HDX kinetic analysis of apoA-I in rsHDL indicates that bimodal HDX kinetics, observed in many of the apoA-I peptides, comes from conformational diversity of apoA-I chains and/or the presence of distinct interaction environments. The distinct dynamic behavior of apoA-I chains in rsHDL suggests that they are configured as dimer/monomer, consistent with the HdHp model of rsHDL.”

On the other hand, the elastic incoherent neutron scattering (EINS) data gives information about the overall dynamics of HDL and not its structure. This data was measured and reported here for the first time and it is therefore not confirmatory.

Reviewer’s critique:

Although very detailed from a technical point of view, it is difficult to understand the key messages and their biological impact.

Response to reviewer’s critique:

We stated in the Conclusion section the biological impact of the dynamic features of HDL on its function:

“… structural changes in VLDL and rHDL (when remodeled into LDL and rsHDL, respectively) seem to be mirrored by changes in particle dynamics. Various plasma enzymes (e.g. LCAT, PLTP, CETP) and cell receptors (e.g. SR-BI) change HDL dynamics as it matures from discoidal to spherical form. It is conceivable that the change in the softness and flexibility of HDL particles during their life span contributes to the selective interaction of the HDL protein components (apoA-I, apoA-II) with these enzymes, which are critical for HDL remodeling and performance of its main physiological function in reverse cholesterol transport. For example, lecithin cholesteryl acyltransferase (LCAT), which esterifies the free cholesterol on the surface of HDL, seems to easily attach to a softer and more flexible discoidal HDL particle by interacting with specific and more flexible domains of apoA-I (e.g. the Solar Flare loop). In contrast, other plasma enzymes, like phospholipid (PLTP) and cholesteryl ester (CETP) transfer proteins, appear to more preferentially interact with a more rigid HDL particle (spherical HDL) in order to accomplish their lipid transfer functions. Thus, the HDL becomes less dynamic only after cholesterol is converted to cholesteryl ester, and this biophysical change in turn impacts its interaction with remodeling partners.”

Reviewer’s critique:

For EINS studies, reconstituted particles were lyophilized and rehydrated at 200 mg/ml. The authors stated that this procedure did not affect particle morphology and homogeneity: data supporting this statement must be provided.

Response to reviewer’s critique:

EINS it is a well-established, unique, and sophisticated technique for studying the dynamics of biomolecular systems and has been successfully used to study the dynamics of other lipoproteins (LDL/VLDL, Mikl et al JACS 133 (2011) 13213, Peters et al, EPJE 40 (2017) 1), membranes, and lipid bilayers (Peters et al, SciRep 7 (2017) 15339). There is a rich body of published literature on using EINS that validates the method in physiologically relevant biological systems. We have used the method based on its uniqueness and successful application to lipoproteins as documented in the literature (Mikl et al JACS 133 (2011) 13213, Peters et al, EPJE 40 (2017) 1). The question of the effect of lyophilization on biological systems was exhaustively studied by J. Perez et al., Biophys. J. 77 (1999) 454, “Evolution of the Internal Dynamics of Two Globular Proteins from Dry Powder to Solution”, where they came to the conclusion that “… from dry powder to complete coverage at 0.4 g/g, the surface side chains progressively acquire the possibility to diffuse locally, thanks to a few molecules of water that offer them several hydrogen-bonding pathways, and that on subsequent hydration, the main effect of water is to improve the rate of these diffuse motions, but without necessarily creating new hydrogen bonds.” Therefore the difference in the systems is thus of quantitative rather than qualitative nature.

Lyophilization of lipoproteins does not change their composition. We haven’t noticed any morphological changes in the lipoprotein samples after lyophilization either before or after the EINS measurements. The samples were weighed and inspected before and after measurements to detect loss of mass or changes in texture, a standard procedure used in the EINS field, thus, the type of dynamics EINS probes (on a picosecond-nanosecond scale) was not altered. We included in the manuscript and the supplementary material all available information about sample preparation, sample monitoring, and sample measurement resulted from the EINS experiments.

Reviewer 3 Report

The manuscript “Protein backbone and average particle dynamics in reconstituted discoidal and spherical HDL probed by hydrogen deuterium exchange and elastic incoherent neutron scattering” by Gogonea et al. examined reconstituted HDL (with only apoA-I) via HDX kinetics. The authors concluded that many apoA-I peptides exhibited bimodal kinetics, and also that the active LCAT binding domain on apoA-I is also bimodal and in a near random coil conformation (as previously postulated by other groups). The manuscript is well written and the data are convincing, despite what appears to be only one batch of rHDL and sHDL being examined. Of very minor comment, change all occurrences of “cholesterol ester” to “cholesteryl ester”.

Author Response

We thank the reviewer for the positive and constructive critiques that helped us improve the manuscript significantly. Bellow we answer reviewer’s critiques one by on:

Reviewer’s critique:

The manuscript “Protein backbone and average particle dynamics in reconstituted discoidal and spherical HDL probed by hydrogen deuterium exchange and elastic incoherent neutron scattering” by Gogonea et al. examined reconstituted HDL (with only apoA-I) via HDX kinetics. The authors concluded that many apoA-I peptides exhibited bimodal kinetics, and also that the active LCAT binding domain on apoA-I is also bimodal and in a near random coil conformation (as previously postulated by other groups). The manuscript is well written and the data are convincing, despite what appears to be only one batch of rHDL and sHDL being examined. Of very minor comment, change all occurrences of “cholesterol ester” to “cholesteryl ester”.

Response to reviewer’s critique:

The reviewer is right, the EINS measurements were performed on a single batch of rHDL/sHDL because neutron beam is a very precious and expensive resource, which is obtained competitively. The EINS measurements required almost two weeks of neutron beam, therefore, we could not afford to measure multiple batches of HDL samples. On the other hand, the hydrogen deuterium exchange mass spectrometry measurements we performed on triplicates of spherical HDL samples.

At reviewer’s suggestion we made this modification within the manuscript.

Round 2

Reviewer 1 Report

No further comment.